# Divergent Morphologies and Common Signaling Features of Active and Inactive Oncogenic *RHOA* Mutants in Yeast

**DOI:** 10.3390/cells14181439

**Published:** 2025-09-15

**Authors:** Chenwei Wang, Shinsuke Ohnuki, Anna Savchenko, Hiroyuki Aburatani, Satoshi Yoshida, Riko Hatakeyama, Yoshikazu Ohya

**Affiliations:** 1Department of Integrated Biosciences, Graduate School of Frontier Sciences, The University of Tokyo, Kashiwa 277-8562, Japan; 7794921937@edu.k.u-tokyo.ac.jp (C.W.); shinsuke.ohnuki@gmail.com (S.O.); msubiochemist@gmail.com (A.S.); 2Department of Cardiology, Cardiovascular Research Institute Maastricht, Maastricht University Medical Center, 6229 ER Maastricht, The Netherlands; 3Genome Science Division, Research Center for Advanced Science and Technology, The University of Tokyo, Tokyo 153-8904, Japan; haburata-tky@umin.ac.jp; 4School of International Liberal Studies, Nishi-Waseda Campus, Waseda University, Tokyo 169-8050, Japan; satosh@waseda.jp; 5Institute of Medical Sciences, School of Medicine, Medical Sciences and Nutrition, University of Aberdeen, Aberdeen AB25 2ZD, UK; 6Department of Science and Technology Innovation, Nagaoka University of Technology, 1603-1 Kamitomioka, Nagaoka 940-2188, Japan; 7Collaborative Research Institute for Innovative Microbiology, The University of Tokyo, Tokyo 113-8657, Japan

**Keywords:** *RHOA*, *RHO1*, cancer, yeast morphology, CalMorph

## Abstract

*RHOA*, a member of the Rho family of small GTPases, harbors recurrent mutations in diverse cancers, but how these mutations cause their cellular effects remains poorly understood. To investigate their cellular consequences, we expressed oncogenic *RHOA* variants (R5Q, G17V, C16R, and A161P) in *Saccharomyces cerevisiae*, substituting for the essential yeast homologue *RHO1*. While the E40Q variant failed to complement *RHO1* deletion, other mutants supported viability and enabled phenotypic characterization. All four variants conferred myriocin resistance, suggesting activation of the membrane stress response pathway, but induced no major changes in growth or caspofungin sensitivity. Using high-dimensional image analysis, we quantified 501 morphological parameters and applied principal component analysis and linear discriminant analysis to determine distinct phenotypic profiles. Gain-of-function (C16R and A161P) and loss-of-function (R5Q and G17V) mutants formed separate morphological clusters, indicating functional divergence. Our yeast model enabled systematic dissection of the functions of *RHOA* mutants and highlighted the utility of morphology-based approaches to characterize context-dependent mechanisms of oncogenesis.

## 1. Introduction

Cancer remains a significant public health challenge and is the second leading cause of death globally [1]. Oncogenes play a pivotal role in cancer pathogenesis, with most driver mutations occurring in key signaling molecules such as kinases and small GTPases [2]. Among these, small GTPases have emerged as frequent targets of oncogenic mutations and are increasingly recognized for their contributions to tumor progression [3,4].

Small GTPases function as molecular switches by cycling between an active GTP-bound and an inactive GDP-bound state to regulate a variety of essential cellular processes, including cell growth, vesicle trafficking, and actin-based cytoskeletal remodeling [5]. Dysregulation of this switching mechanism can disrupt proliferation and survival control, contributing to tumorigenesis [6,7]. As a prominent subgroup of small GTPases, the Rho family merits particular attention [8].

The Rho family of small GTPases, a subset of the Ras superfamily conserved across eukaryotes [9], controls cell morphology, motility, adhesion, and migration through actin cytoskeleton dynamics [10,11]. Although RhoA is mutated less frequently than Ras genes in cancer cells, it has been shown to be altered in certain malignancies, including angioimmunoblastic T-cell lymphoma (AITL) and adult T-cell leukemia/lymphoma (ATL), highlighting its context-dependent oncogenic potential [12,13].

Several recurrent point mutations in *RHOA* have been identified across human cancers, with functional consequences that differ by the mutation site and cancer type [12,14]. Notable hotspot mutations include R5Q, G17V, C16R, A161P, and E40Q. R5Q, which substitutes Arg5 with Gln, recurs in diffuse-type gastric carcinoma and Burkitt lymphoma [15,16]. G17V, present in ~67% of AITL cases and other peripheral T-cell lymphomas, abolishes GTP binding ability [16,17]. C16R and A161P are predominantly seen in ATL [18], whereas E40Q occurs in solid tumors such as breast cancer and head-and-neck squamous cell carcinoma [19]. These findings suggest that *RHOA* mutations contribute to oncogenesis across a diverse spectrum of cancers, but likely via distinct mechanisms depending on the mutation type and cellular context.

These *RHOA* mutations can be broadly classified into two functional categories: gain-of-function (dominant active) and loss-of-function. Dominant-negative denotes a mechanism that interferes with wild-type signaling and is not synonymous with loss-of-function. Gain-of-function mutations, exemplified by C16R and A161P, accelerate GTP/GDP cycling, elevate downstream signaling, and promote actin fiber assembly in ATL [14,18,20]. By contrast, loss-of-function mutations, including G17V (dominant-negative) and R5Q (attenuated-output), impair RhoA function. The G17V mutation disrupts GTP binding, rendering the protein inactive and functioning as a dominant-negative driver in AITL and peripheral T-cell lymphomas [17,21,22]. On the other hand, R5Q reduces RhoA activation and has been implicated in Burkitt lymphoma and diffuse large B-cell lymphoma [16,23]. Despite these functional classifications, the downstream mechanisms by which each mutation type promotes oncogenesis remain poorly understood.

In mammalian cells, oncogenic *RHOA* variants have been reported to alter actin organization and cell morphology in a context-dependent manner, with gain-of-function alleles often enhancing RhoA signaling and cytoskeletal remodeling, whereas loss-of-function alleles—including dominant-negative variants—tend to reduce these processes [14,18,20]. However, direct, cross-allele, quantitative comparisons under a single experimental framework remain scarce [3].

To better understand the downstream mechanisms by which *RHOA* mutations drive oncogenesis, we used *Saccharomyces cerevisiae* as a model eukaryotic system. Yeast has proven to be a valuable tool for dissecting conserved signaling pathways, allowing the functional characterization of human disease-associated genes in a genetically tractable context [24,25,26,27]. In this study, various *RHOA* mutant alleles were heterologously expressed in yeast and their effects on growth phenotypes and cellular morphology were assessed. Subtle morphological changes were captured using CalMorph, a high-throughput image analysis pipeline that we developed previously [28]. This approach enabled us to systematically compare the phenotypic consequences of different *RHOA* mutations and to gain insight into the distinct cellular programs they activate, thus providing a novel perspective on their context-dependent oncogenic mechanisms.

## 2. Materials and Methods

### 2.1. Strains and Media

The yeast strains used and constructed in this study are listed in Appendix A. Yeast strains were cultured in Yeast Extract Peptone Dextrose Medium (YPD) consisting of 1% Bacto-Yeast Extract (Thermo Fisher Scientific, Waltham, MA, USA), 2% Bacto-peptone (Thermo Fisher Scientific), and 2% glucose (Wako Chemicals, Osaka, Japan), YPD supplemented with 1 M sorbitol, or Synthetic Dextrose Medium Containing Casamino Acids Medium (SDCA) composed of 0.17% yeast nitrogen base without amino acids and ammonium sulfate (BD Biosciences, Franklin Lakes, NJ, USA), 0.5% ammonium sulfate (Wako Chemicals), 0.5% casamino acids (Thermo Fisher Scientific), and 2% glucose (Wako Chemicals) at 30 °C. YPD plus hygromycin B (300 μg/mL) and YPD plus clonNAT (100 μg/mL) media were used for strain selection. 5-FOA medium consisting of 0.1% 5-fluorotic acid (Toronto Research Chemicals, Toronto, ON, Canada), 0.67% yeast nitrogen base without amino acids and ammonium sulfate (BD Biosciences), 0.5% casamino acids, and 2% glucose (Wako Chemicals) was used for counterselection of yeast strains containing a *URA3* marker plasmid.

Unless otherwise noted, morphological analyses were conducted under standard, drug-free growth conditions to permit unbiased comparisons across strains. Myriocin and caspofungin were not present during image acquisition; their effects were evaluated separately by growth-based assays.

### 2.2. Transformation, Mating, and Screening

To generate self-fluorescent strains, two plasmids encoding fluorescently tagged proteins (pKN4 and pKN23) were constructed and transformed into yeast cells. pKN4 contained an HTA2–3× mRuby gene (for nuclear labeling) and a hygromycin B resistance (HygR) gene. pKN23 contained a Lifeact–mNeonGreen gene (for actin labeling) and a clonNAT resistance gene. The obtained yeast strains were mated on YPD media and diploid strains were selected on YPD medium with both antibiotics (YPD plus 300 μg/mL hygromycin B and 100 μg/mL clonNAT). The *URA3*-based *RHO1* plasmid was eliminated using 5-FOA medium and loss of the plasmid was confirmed using SD medium lacking uracil.

To enable rapid identification of correct integrants at the *ADE3* locus, we used an *ade2* background with *ADE3* as a color marker: under adenine-limiting conditions, *ade2 ADE3* colonies appear red, whereas *ade2 ade3Δ::insert* colonies are white due to lack of pigment accumulation. Putative white colonies were restreaked and confirmed by colony PCR to verify *ADE3*–locus integration.

### 2.3. Colony PCR

To validate the integration of desired fragments, the genomes of yeast transformants were subjected to colony polymerase chain reaction (PCR) using the primers listed in Appendix A.

### 2.4. Cell Wall Fluorescence Staining, Microscopy, and Image Processing

Strains were cultured to early log phase (1 × 10^6^ to 2 × 10^6^ cells/mL) for staining with CF^®^350–Concanavalin A (Con A). Cells were collected by centrifugation and resuspended in Hanks’ Balanced Salt Solution (HBSS) with calcium and magnesium containing 200 µg/mL CF^®^350–Con A. Samples were incubated at room temperature for 10 min, washed five times with HBSS, and then observed using an Axio Imager microscope (100/100 light intensity, reflected light aperture diaphragm at maximum) equipped with a 100× EC Plan-Neofluar lens (Carl Zeiss, Oberkochen, Germany), a CoolSNAP HQ cooled charge-coupled device camera (Roper Scientific Photometrics, Tucson, AZ, USA), and AxioVision software v4.5 (Carl Zeiss). Images were processed with CalMorph v1.3 image analysis software, designed for diploid yeast, to quantify cell wall, actin, and nuclear morphology [29]. Five biological replicates were analyzed per strain. Images with segmentation errors or failed feature extraction were excluded from downstream analyses.

### 2.5. Data Normalization

Statistical analyses were performed in R (R Core Team, Vienna, Austria). All 501 morphological features were Z-score normalized for each replicate, using the *RHOA*/*rhoAΔ* strain as the null distribution, before statistical analysis. The dispersion of the null distribution for each feature was estimated after model selection by the Akaike Information Criterion (AIC), choosing between equal-variance and unequal-variance models across strains. Average Z-scores for each strain were used in principal component analysis (PCA).

### 2.6. Principal Component Analysis

PCA was conducted to reduce dimensionality and identify global morphological variation. The first PCA was performed using the correlation matrix of averaged Z-scores across all strains to maximize morphological differences among them. Z-scores for each replicate were subsequently projected onto the PC axes using the estimated eigenvalues, and the resulting PC scores were used to calculate Euclidean distances.

For linear discriminant analysis (LDA), Z-scores from each replicate were subjected to PCA after removing one morphological parameter (ACV104_C) across all strains and one replicate from the *RHOA*/*rhoAΔ* strain due to missing values. A secondary PCA was performed on subsets of traits significantly associated with the LDA axes, using data from 114 wild-type replicates as the null distribution, to refine interpretation of mutant-specific features [28]. Another PCA was also conducted on all 501 traits using the 114 wild-type replicates to maintain an orthogonal morphological space, where PC scores projected from Z-scores were used in correlation analyses and hierarchical clustering.

### 2.7. Hierarchical Clustering

Morphological similarity among strains was assessed using Pearson’s correlation coefficients calculated from mean PC score profiles. Strains were grouped by hierarchical clustering with average-linkage aggregation, and the resulting dendrogram was used to evaluate phenotypic similarity.

### 2.8. Generalized Linear Model Analysis

To identify traits altered by each *RHOA* mutation, generalized linear models with a one-way ANOVA design were fitted to all 501 features, comparing each variant against the control (*RHOA*/*rhoAΔ*). The significance of model fitting and the difference from the control were determined sequentially using likelihood ratio tests and Wald tests, respectively, with multiple testing correction by Storey’s false discovery rate (FDR). Features with FDR < 0.05 were considered significantly fitted by the model or significantly different from the control.

### 2.9. Linear Discriminant Analysis

To evaluate whether *RHOA* variants could be distinguished based on morphological profiles, LDA was conducted on the dimensionally reduced data set. The input data consisted of the top 17 principal components (PCs) derived from Z-scores of the full morphological set, excluding one morphological feature across all strains and one replicate in *RHOA*/*rhoAΔ* with missing values. These components accounted for >90% of the total variance. The resulting linear discriminants revealed clear separation among control, gain-of-function-variant, and loss-of-function-variant groups.

### 2.10. Feature Selection and Interpretation

Morphological traits significantly correlated with either the first (LD1) or second (LD2) linear discriminant (FDR < 0.1) were retained for further interpretation. These traits were subjected to a secondary PCA to visualize underlying patterns, and their biological relevance was interpreted based on known CalMorph categories. To summarize phenotypic differences, representative cell schematics were created to illustrate characteristic features of each variant class across cell cycle stages.

### 2.11. Cell Lysate Preparation and Immunoblotting Analysis

Cells in mid-log phase were treated with trichloroacetic acid to a final concentration of 6.7% (*w*/*v*), pelleted, washed with ice-cold 70% ethanol, dried, dissolved in urea buffer consisting of 50 mM Tris-HCl (pH 8.0), 5 mM EDTA, 6 M urea, 1% sodium dodecyl sulfate (SDS), 1 mM Pefabloc, and PhosSTOP (one tablet per 10 mL), and disrupted with glass beads using a FastPrep-24 homogenizer (MP Biomedicals, Irvine, CA, USA). Samples were heated at 65 °C for 10 min, and then again after addition of Laemmli SDS sample buffer. Proteins were separated by SDS–polyacrylamide gel electrophoresis (SDS–PAGE) and analyzed by immunoblotting with anti-phospho-p44/42 MAPK (Erk1/2) (Thr202/Tyr204) antibody (#4370; Cell Signaling Technology, Danvers, MA, USA).

## 3. Results

### 3.1. Construction of Phenotypic Assay Strains Expressing Oncogenic RHOA Variants in Yeast

To investigate the intracellular functions of oncogenic *RHOA* mutations, we constructed a panel of *S. cerevisiae* strains expressing either the wild-type human *RHOA* or one of four point-mutant alleles (R5Q, G17V, C16R, and A161P) in place of yeast *RHO1* (see “Section 2”) (Figure 1A). In these diploids, both chromosomal copies of the essential *RHO1* were deleted, and complementation was provided by plasmid-borne human *RHOA*. We then used a 5-FOA plasmid-shuffle assay to test complementation of essential *RHO1* function: constructs that complement support growth on 5-FOA, whereas non-complementing alleles fail to grow (Figure 1B). We also evaluated the allele E40Q; upon *URA3*-plasmid eviction on 5-FOA, no viable colonies were recovered, rendering its terminal phenotype indistinguishable from the null under our conditions. Accordingly, E40Q was excluded from downstream morphological profiling (Figure 1B).

To visualize cellular structures, two fluorescent marker constructs were integrated into the chromosomes: pKN23 (Lifeact–mNeonGreen & clonNAT) for labeling the actin cytoskeleton and pKN4 (HTA2–3× mRuby & HygR) for labeling nuclei. Upon staining with the cell wall-specific dye CF^®^350-Con A, the cell wall, actin, and nucleus were clearly visualized by fluorescence microscopy (Figure 1C), confirming that these strains were suitable for quantitative morphological profiling.

### 3.2. Growth Phenotypes of RHOA Mutants

To evaluate the effects of *RHOA* mutations on yeast growth, phenotypic assay strains were cultured under various conditions. Temperature sensitivity assays revealed that most *RHOA* variants exhibited comparable or only slightly reduced growth relative to the wild-type controls, with no apparent sensitivity to low or high temperatures (Figure 2A).

We next assessed responses to myriocin and caspofungin. Myriocin inhibits serine palmitoyltransferase, the first enzyme in sphingolipid biosynthesis [30,31]. Yeast copes with the myriocin-triggered membrane stress by activating the TORC2 signaling pathway, which operates upstream of, and/or in parallel with, Rho1 [32]. Spot assays demonstrated that all strains expressing *RHOA* mutants (R5Q, G17V, C16R, and A161P) showed increased resistance to myriocin compared to the *RHOA/RHOA* and *RHOA*/*rhoAΔ* controls (Figure 2B), suggesting that these mutations may activate the TORC2 pathway in yeast.

Caspofungin, an inhibitor of 1,3-β-glucan synthase (a known Rho1 target) [33], was used to assess the Rho1-like activity of the *RHOA* variants. No clear differences in caspofungin sensitivity were observed among the *RHOA* mutants (Figure 2C), although slight variations were detected at low concentrations (5–50 ng/mL) (Appendix A). In contrast, the *RHO1*/*RHO1 ade3* strain showed increased sensitivity at higher caspofungin concentrations, likely due to reduced Rho1 protein expression (Appendix A).

Given that Rho1 also regulates the cell wall integrity (CWI) pathway, we examined whether *RHOA* variants alter this pathway by analyzing the phosphorylation status of Slt2 [34], a MAP kinase in the CWI pathway. Slt2 phosphorylation is known to be increased in *RHO1*-null strains due to cell wall defects [35]. To assess whether oncogenic *RHOA* mutations elicit a similar response, we measured Slt2 phosphorylation in strains expressing the variants. Among the phenotypic assay strains expressing oncogenic *RHOA* variants, Slt2 phosphorylation varied across biological replicates (*n* = 3). After normalization to total Slt2 and a loading control, only A161P reproducibly exhibited elevated Slt2 phosphorylation relative to the *RHOA/RHOA* control; C16R showed occasional, non-reproducible increases, whereas G17V and R5Q were indistinguishable from controls (Figure 2D). We therefore interpret these immunoblot data as evidence for allele-specific activation (A161P) rather than a uniform change across variants. Under *RHO1*-replete conditions, however, the mutations had minimal effects (Appendix A). Collectively, these observations suggest that, in the presence of Rho1, the *RHOA* mutants share a largely common phenotypic profile.

### 3.3. Morphological Profiling of RHOA Mutants

To investigate the morphological consequences of *RHOA* mutations, we conducted quantitative morphological analyses using phenotypic assay strains. Fluorescence images of the cell wall, actin cytoskeleton, and nucleus were acquired from five biological replicates for four heterozygous *RHOA* mutant strains (*RHOA*/*C16R*, *RHOA*/*A161P*, *RHOA*/*G17V*, and *RHOA*/*R5Q*) and four control strains (*RHO1*/*RHO1*, *RHO1*/*RHO1 ade3*, *RHOA*/*RHOA*, and *RHOA*/*rhoAΔ*) (Figure 3). Morphological data sets comprising 501 parameters were extracted from images of a minimum of 200 cells per replicate using the yeast-specific image analysis software CalMorph [28,29].

Morphological deviations were quantified by calculating Euclidean distances [36] from the wild-type diploid reference (*RHO1*/*RHO1*) in PC space. On PCA using standardized Z-scores, six PCs accounted for over 90% of the total variance (Appendix A). The heterozygous *RHOA* mutants exhibited substantially elevated morphological abnormality scores relative to both *RHO1*/*RHO1* and *RHOA*/*rhoAΔ* (Figure 4A), indicating that these mutants have distinct morphological phenotypes.

To assess the relations among morphological profiles, we next examined pairwise correlations of PC scores [36]. For robust comparisons, 100 PCs cumulatively explaining more than 99% of the variance were derived from 114 biological replicates of the standard strain BY4743 (Appendix A). Pairwise correlation coefficients among the heterozygous mutants were comparatively high (0.649–0.842), whereas correlations between heterozygous *RHOA*/*rhoAΔ* and each mutant were notably lower (0.221–0.271) (Figure 4B). Correlations between homozygous *RHOA*/*RHOA* and each mutant were also low (0.212–0.469). Hierarchical clustering based on Pearson’s correlation grouped three heterozygous mutants (A161P, C16R, and G17V) into a single cluster (Figure 4C). Although the two loss-of-function alleles (G17V and R5Q) did not form a discrete cluster, they were more similar to each other than to *RHOA*/*rhoAΔ*. Visualization in two-dimensional PC space further substantiated the separation between *RHOA*/*rhoAΔ* and the heterozygous mutants (Figure 4D). Therefore, PCA and correlation analysis clearly demonstrated a morphological separation between *RHOA*/*rhoAΔ* and the heterozygous *RHOA* oncogenic mutants.

Notably, A161P/C16R (gain-of-function) and G17V (loss-of-function, dominant-negative) clustered together despite their opposite biochemical mechanisms, indicating convergent morphological outputs along actin/cell-wall axes. In contrast, R5Q—an attenuated-output (loss-of-function, non-dominant-negative) allele—clustered with *RHO1*/*RHO1*, consistent with its milder overall deviations. Thus, the clustering reflects the magnitude and pattern of downstream perturbation rather than the simple gain-of-function/loss-of-function dichotomy (Figure 4C).

### 3.4. Differential Morphological Effects of Gain- and Loss-of-Function Mutants

To further dissect the phenotypic distinctions among *RHOA* variants, we focused on morphological parameters that showed significant differences in *RHOA* heterozygous mutants. Using a generalized linear model combined with one-way ANOVA, we identified 164 parameters that exhibited significant changes across the four mutants (FDR < 0.05, Wald test with Storey’s correction) (Appendix A, Appendix A). Among these, A161P showed the greatest number of altered parameters. Based on CalMorph IDs, these significant parameters comprised nuclear (D*) (70/164; 42.7%), actin (A*) (51/164; 31.1%), and cell-shape (C*) (43/164; 26.2%).

Next, we performed linear discriminant analysis (LDA) [37] on 500 parameters (from 501 features; one excluded due to missing values), after reducing the dataset to 17 PCs explaining more than 90% of the variance (Appendix A). This analysis clearly separated the five strains (four *RHOA* heterozygotes and *RHOA*/*rhoAΔ*) into three distinct classes (Figure 5A). The first discriminant axis (LD1) differentiated *RHOA* heterozygotes from *RHOA*/*rhoAΔ*, while the second axis (LD2) distinguished gain-of-function from loss-of-function mutants. The largest loadings were as follows: for LD1, the proportion of actin at the bud neck (A9_A1B) and cell length (C103_A1B); for LD2, the variability in actin patch brightness (ACV122_C) (Figure 5B,C).

Finally, Figure 6 summarizes class-level cartoons and highlights CalMorph features loading on LD1/LD2 that distinguish gain-of-function (C16R and A161P) and loss-of-function (R5Q and G17V) from control. These morphometric profiles highlight the traits distinguishing functional classes of *RHOA* oncogenic mutants and provide insights into their intracellular impacts.

## 4. Discussion

We established a yeast-based system for systematic phenotypic analysis of cancer-associated *RHOA* mutations. Growth assays combined with high-dimensional morphological profiling delineated both similarities and differences between gain-of-function (C16R and A161P) and loss-of-function (R5Q and G17V) variants. Although the mutants showed comparable sensitivities to myriocin and caspofungin, morphological analysis revealed clear distinctions between the two functional classes, indicating that these *RHOA* variants engage distinct cellular programs even in a simplified eukaryotic context.

All four variants conferred resistance to myriocin, a trigger of TORC2 activation, suggesting activation of TORC2 signaling through alternative routes. This observation was consistent with reports that GDP-bound RhoA can modulate TORC2 via noncanonical mechanisms [38] and may explain the shared cytoskeletal and survival responses despite divergent GTPase activity. In contrast, caspofungin sensitivity assays showed no consistent patterns of resistance or sensitivity. Furthermore, activation of the CWI pathway was not detected when RHO1 was present, indicating that *RHOA* variants do not activate CWI signaling in the presence of endogenous Rho1. Taken together, these findings suggested that *RHOA* mutations primarily modulate the TORC2–Pkc1 signaling axis rather than Fks1-dependent glucan synthesis targeted by caspofungin [39,40,41].

While our data demonstrated similarities in growth phenotypes among *RHOA* mutants, their underlying molecular mechanisms remain to be elucidated. A central question is whether TORC2 and Pkc1 act in parallel or sequentially downstream of these variants. The robust myriocin resistance observed across all mutants suggested convergence on TORC2 signaling, but the extents to which this led to Pkc1 activation and CWI engagement remain unclear. Although A161P reproducibly elevated Slt2 phosphorylation, the class separation in morphology persisted without consistent MAPK changes, indicating contributions from Rho1-dependent pathways parallel to Pkc1–Slt2 (e.g., formins/actin organization, TORC2–Ypk1/2). Genetic epistasis analysis and use of TORC2-specific reporters in yeast may clarify this signaling hierarchy.

Quantitative morphological profiling was the most effective discriminator of *RHOA* variant classes. PCA and LDA analyses clearly separated *RHOA*/*rhoAΔ*, gain-of-function, and loss-of-function strains. Features driving this separation included actin bud neck localization, cell length, and variability in actin patch brightness, suggesting that actin cytoskeleton remodeling acts as a key downstream effector of *RHOA* mutations [42]. These results suggested that even catalytically impaired *RHOA* variants engage cytoskeletal pathways through distinct mechanisms. Together, these orthogonal phenotypes support mechanistic divergence between gain-of-function and loss-of-function classes, even when coarse growth/drug readouts converge.

Notably, while mammalian studies have linked *RHOA* oncogenic variants to cytoskeletal and morphological changes [14,18,20], a unified, quantitative separation of multiple alleles under matched conditions has not been firmly established [3]. Our yeast-based phenomics clearly separates gain-of-function and loss-of-function classes along actin-related features, providing a complementary reference and underscoring the need for similarly standardized, quantitative phenotyping in mammalian systems.

Finally, the morphological features identified here provide a robust means of screening chemical or genetic modifiers that selectively affect gain- or loss-of-function RHOA signaling. Such modifiers could help to identify pathways uniquely engaged by each class of mutation and ultimately inform therapeutic strategies for *RHOA*-driven cancers.

## 5. Conclusions

The yeast-based platform integrating growth assays and high-dimensional morphological profiling used in this study were effective for characterizing the functional classes of cancer-associated *RHOA* mutations. These phenotypic profiles support the translational relevance of our system and may inform future studies in clinical and mammalian contexts. Our findings highlight the value of yeast functional genomics for dissecting the mechanistic diversity of small GTPase mutations and underscore the utility of morphometric profiling as a scalable approach for functional annotation in cancer genomics. Future studies employing genetic and biochemical approaches will further clarify how specific *RHOA* mutations engage divergent signaling networks and provide insights into their oncogenic potential and therapeutic targeting.

## Figures and Tables

**Figure 1 cells-14-01439-f001:**
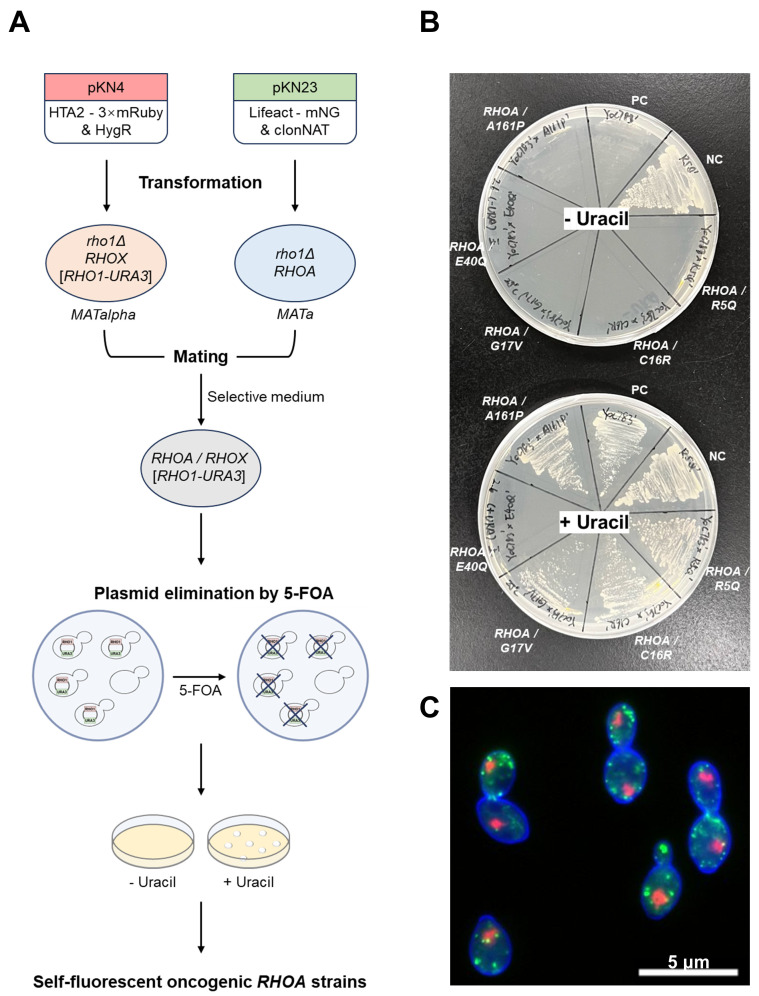
Construction and validation of self-fluorescent yeast strains expressing oncogenic *RHOA* variants. (**A**) Workflow for constructing self-fluorescent humanized strains expressing oncogenic *RHOA* variants. Haploid *MATa* and *MATα* yeast strains lacking the essential *RHO1* gene and harboring a *RHO1-URA3* rescue plasmid were transformed with plasmids pKN4 (HTA2–3× mRuby and HygR) and pKN23 (Lifeact–mNeonGreen and clonNAT), respectively. After transformation, the two parental strains were mated, and diploids were selected on media containing both antibiotics. The rescue plasmid was eliminated by 5-FOA selection. Successful plasmid loss was verified by growth on SD plates with and without uracil. The resulting strains expressed nuclear and actin markers and served as the base for the following quantitative morphological profiling. (**B**) Plasmid-shuffle assay. Cells carrying a *URA3*-marked *RHO1* plasmid and an LEU2-marked test plasmid were spotted on SC + 5-FOA (counter-selection) to evict the *URA3* plasmid. Growth on 5-FOA indicates complementation of essential Rho1 function by the test plasmid; no growth indicates failure to complement. A positive control (PC; no *URA3*) and a negative control (NC; with *URA3*) validated the selection system. *RHOA*/A161P, *RHOA*/G17V, *RHOA*/C16R, and *RHOA*/R5Q grew on 5-FOA, indicating complementation. *RHOA*/E40Q failed to grow, consistent with non-complementation. (**C**) Representative fluorescence microscopy image of diploid yeast strain expressing nuclear (red) and actin (green) markers. The cell wall was stained using the fixative-free dye CF^®^350-ConA (blue). Scale bar: 5 µm.

**Figure 2 cells-14-01439-f002:**
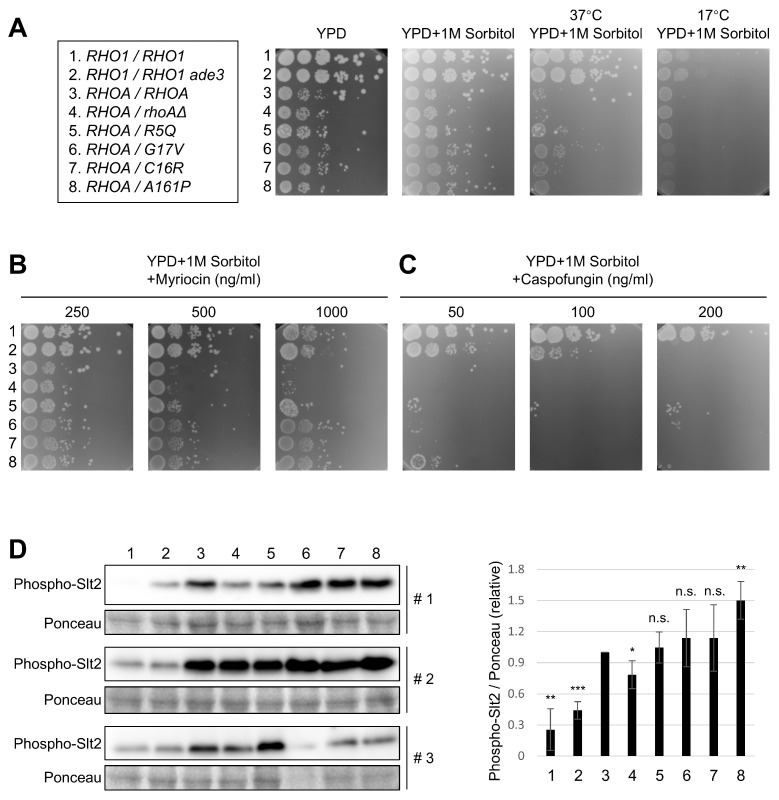
Growth phenotypes and CWI signaling response of *RHOA* variants under stress conditions. (**A**) Growth phenotypes of *RHOA* variants under various temperatures. Variants or controls were grown on YPD + 1 M sorbitol medium at 37 °C (heat) and 17 °C (cold). Variants showed modest differences compared to controls under these conditions. Images were taken after 2 days of incubation. (**B**) Growth phenotypes of *RHOA* variants in the presence of myriocin. Spotting assays on YPD + 1 M sorbitol supplemented with 250–1000 ng/mL myriocin showed strong resistance in all *RHOA* variants comparable to the *RHOA*/*RHOA* and *RHOA*/*rhoAΔ* control strains. Images were taken after 2 days of incubation. (**C**) Growth phenotypes of *RHOA* variants in the presence of high-dose caspofungin. Spotting assays on YPD + 1 M sorbitol with 50–200 ng/mL caspofungin showed no differences in colony growth across strains. Images were taken after 2 days of incubation. (**D**) Western blotting analysis and quantification of Slt2 MAPK phosphorylation in *RHOA* variants lacking *RHO1*. The intensity of phospho-Slt2 was normalized relative to Ponceau staining and plotted relative to the *RHOA*/*RHOA* control (lane 3). Bar graphs represent the mean ± SD from three replicates. Statistical comparisons were made against *RHOA*/*RHOA* using the unpaired two-tailed *t*-test (*p* < 0.05). ***** *p* < 0.05; ****** *p* < 0.01; ******* *p* < 0.001; n.s., not significant.

**Figure 3 cells-14-01439-f003:**
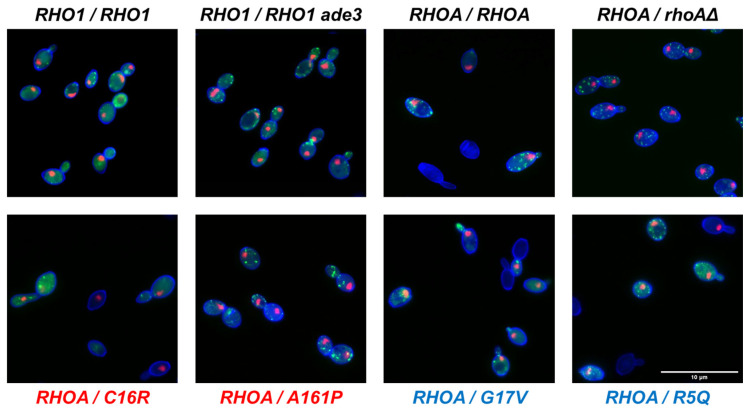
Morphological phenotypes of the *RHOA* strains. Actin (green), nucleus (red), and cell wall (blue) signals are shown in merged images. Red stands for gain-of-function variants and blue for loss-of-function variants. Scale bar: 10 μm.

**Figure 4 cells-14-01439-f004:**
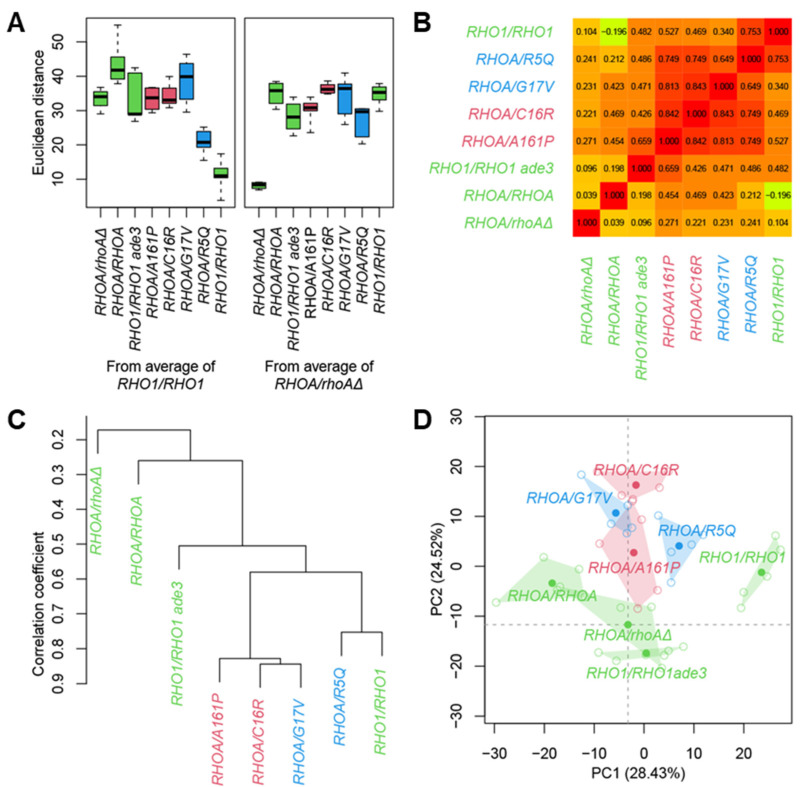
PCA and clustering analysis of *RHOA* variants. (**A**) Morphological abnormality scores of *RHOA* heterozygous mutants based on Euclidean distances in PC space. Box plots show Euclidean distances of various *RHOA* heterozygous mutants from the centroids of either *RHO1*/*RHO1* (left panel) or *RHOA*/*rhoAΔ* (right panel). (**B**) Morphological similarity matrix based on Pearson’s correlation coefficients among mean Z-score profiles of each strain. The table quantifies global phenotypic resemblance among *RHOA* variants and the control. (**C**) Hierarchical clustering dendrogram based on Pearson’s correlations among mean morphological profiles. (**D**) Scatter plot of PCA showing PC1-PC2. Filled circles indicate the PC scores based on the mean Z-values of each strain, while open circles represent individual biological replicates. For visualization purposes, missing values in Z-scores of each replicate were interpolated by substituting zero and projected onto PC axes retained by mean Z-values. Polygons enclose the replicates of each strain.

**Figure 5 cells-14-01439-f005:**
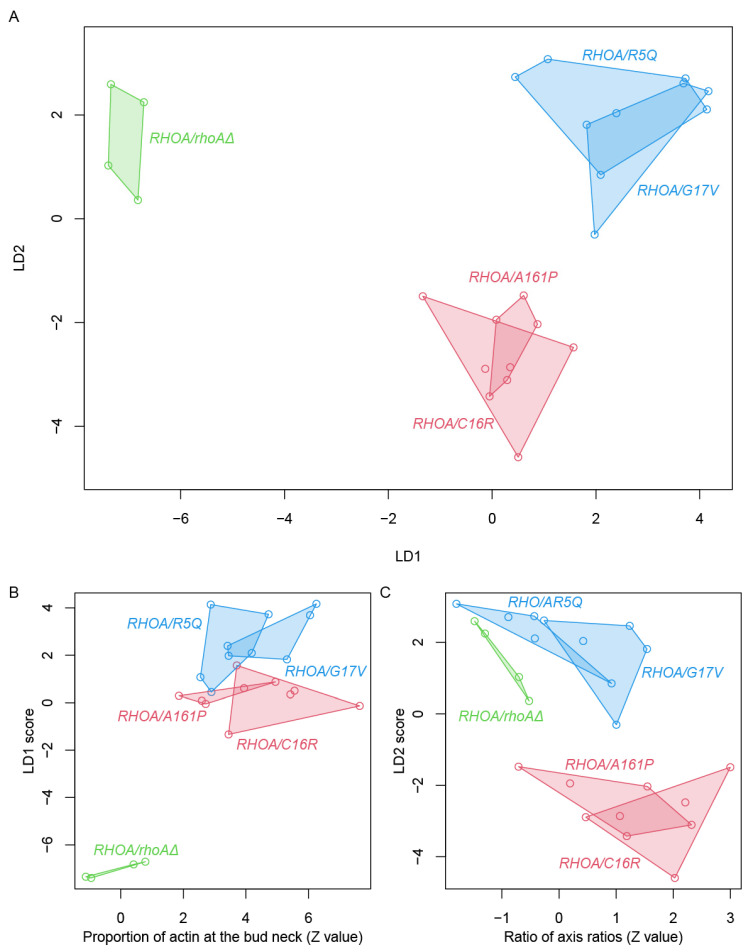
LDA revealed morphological discriminants of RHOA variant classes. (**A**) LDA plot showing LD1–LD2 using the top 17 PCs from PCA. Each dot represents one biological replicate. Strains clustered into three groups. Clusters are outlined with polygons. (**B**) Representative feature highly correlated with LD1: the proportion of actin at the bud neck (A9_A1B), which separates *RHOA*/*rhoAΔ* from oncogenic *RHOA* heterozygotes. (**C**) Representative feature associated with LD2: the ratio of axis ratio (the ratio of bud to mother cell in the ratio of long axis to short axis length, C116_C), which distinguishes gain- from loss-of-function mutants.

**Figure 6 cells-14-01439-f006:**
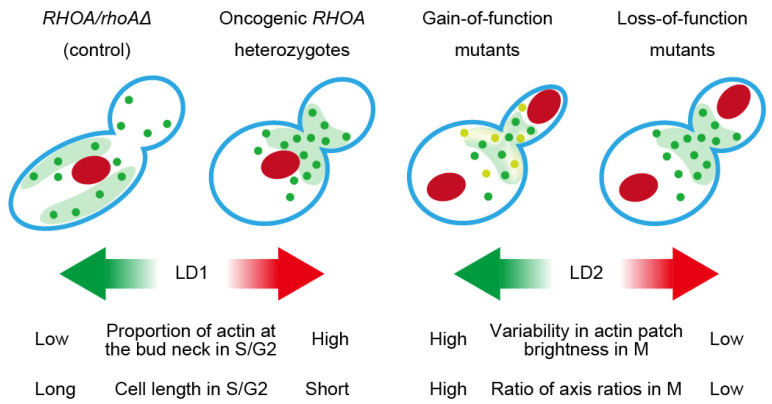
Class-level schematic of morphological signatures and discriminant axes. Cartoons summarize control (*RHOA*/*rhoAΔ*) and oncogenic *RHOA* heterozygotes (gain-of-function and loss-of-function mutants) across cell cycle phases (S/G2, M). Representative features contributing to LD axes (LD1 and LD2) are listed with CalMorph IDs—e.g., proportion of actin at the bud neck in S/G2 phase (A9_A1B), cell length in S/G2 phase (C103_A1B, long-axis length in mother), variability of actin at the bud neck in M phase (ACV122_C), ratio of axis ratios in M phase (C116_C). Colors: green dots, actin patches; light green, actin-rich region; red, nucleus; red and green arrows, increase and decrease in LD scores in Figure 5. See Appendix A for full feature definitions.

## Data Availability

The data presented in this study are available on reasonable request from the corresponding author.

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
