# Peer review of "Divergent Morphologies and Common Signaling Features of Active and Inactive Oncogenic RHOA Mutants in Yeast"

_cells, 2025, doi:10.3390/cells14181439_

Round 1

Reviewer 1 Report

Comments and Suggestions for Authors

Review of “Divergent Morphologies and Common Signaling Features of Active and Inactive Oncogenic RHOA Mutants in Yeast” by Wang et al.

Proteins that control cell signaling and polarity are important for many biological processes, and one of the best characterized are G-proteins. Monomeric G-proteins of the Ras, Rho, and Cdc42 families are critical for the organization of cell shape, cell motility, actin organization, and signaling. Due to their importance, monomeric G-proteins when misregulated are typically associated with cancer and other diseases. Alleles that impact G-protein function occur in many types of cancers; however, due to the complexities of working with mammalian cells – and misregulatred cancer cells - understanding the output phenotypes of these alleles can be complicated. Using a reductionist approach, alleles of the human RHOA protein were moved into yeast, replacing the existing homologous Rho1 gene. The utility of the yeast system comes from its relative simplicity, from the uniformity of cellular phenotypes in large populations, and from the powerful genetic, molecular, and cell biological tools available. In this study, a set of morphological parameters in yeast were examined for phenotypes resulting from RHOA mutations associated with human cancers. Gain- and loss-of-function alleles were examined as well as heterozygosity. A large amount of phenotypic data was presented in a clear and simple manner. Powerful tools previously developed by the laboratory (e.g., CalMorph) allowed clear phenotypic assignments. The utility of morphological analyses was touted as a way of understanding RHO function and biology. The study was carefully conducted. The experiments were well controlled. Appropriate statistical analysis was used. The paper was carefully written. Several pieces of information would benefit from additional clarification. A few questions might be helpful to consider for the readers.

Questions for the authors:

  1. Are the alleles known to impact mammalian cell morphologies? There was some mention of this in the introduction, but it might be helpful to compare GOF and DomNeg phenotypes across species.

  1. Is the actin cytoskeleton perturbed in these mutants? Were there any patterns of interest?

  1. Are dominant negative mutations the same as loss-of-function mutations? Perhaps the labelling of these alleles could be considered?

  1. Did the terminal phenotype of the E40Q mutant differ from the null strain?

  1. The labeling of Figure 1B is confusing, because none of the mutants appear to grow on media lacking uracil. However, the text states the plasmids complement the loss of the control (URA+) plasmid. I think the text and labelling of the figure (e.g., state that the plates contain FOA) might help the reader.

  1. What is the significance of ade3? This was not introduced.

  1. When the mouse hovers over the figures, the text is in Japanese.

  1. Why are the P~Slt2 replicates so different? Do differences in MAPK underlie the morphological differences? If not, then what might cause the phenotypic differences?

  1. Were the morphological analyses examined under conditions where growth was impacted by the drugs tested?

Author Response

Reviewer #1:

Proteins that control cell signaling and polarity are important for many biological processes, and one of the best characterized are G-proteins. Monomeric G-proteins of the Ras, Rho, and Cdc42 families are critical for the organization of cell shape, cell motility, actin organization, and signaling. Due to their importance, monomeric G-proteins when misregulated are typically associated with cancer and other diseases. Alleles that impact G-protein function occur in many types of cancers; however, due to the complexities of working with mammalian cells – and misregulatred cancer cells - understanding the output phenotypes of these alleles can be complicated. Using a reductionist approach, alleles of the human RHOA protein were moved into yeast, replacing the existing homologous Rho1 gene. The utility of the yeast system comes from its relative simplicity, from the uniformity of cellular phenotypes in large populations, and from the powerful genetic, molecular, and cell biological tools available. In this study, a set of morphological parameters in yeast were examined for phenotypes resulting from RHOA mutations associated with human cancers. Gain- and loss-of-function alleles were examined as well as heterozygosity. A large amount of phenotypic data was presented in a clear and simple manner. Powerful tools previously developed by the laboratory (e.g., CalMorph) allowed clear phenotypic assignments. The utility of morphological analyses was touted as a way of understanding RHO function and biology. The study was carefully conducted. The experiments were well controlled. Appropriate statistical analysis was used. The paper was carefully written. Several pieces of information would benefit from additional clarification. A few questions might be helpful to consider for the readers.

Response: Thank you for your thoughtful and generous evaluation of our work. We appreciate your recognition of the rationale for using a reductionist yeast system to study oncogenic RHOA alleles, the breadth and clarity of the morphological dataset, the utility of our CalMorph pipeline, and the care and statistical rigor of the experiments. We agree that several aspects would benefit from additional clarification; in the revision we have expanded the relevant sections and address each point in our point-by-point responses. Your constructive feedback has helped us sharpen the manuscript and better convey its broader significance.

Questions for the authors:

1. Are the alleles known to impact mammalian cell morphologies? There was some mention of this in the introduction, but it might be helpful to compare GOF and DomNeg phenotypes across species.

Response 1: We thank the reviewer for this helpful suggestion. We have revised the Introduction to briefly summarize reported cytoskeletal/morphological effects of oncogenic RHOA variants in mammalian systems and to clarify how these relate to our yeast findings. We also added a short paragraph in the Discussion that explicitly compares the separation we observe between gain-of-function and loss-of-function (dominant-negative) classes in yeast with what has been described in mammalian cells. To our knowledge, a side-by-side, quantitative comparison across multiple RHOA alleles under uniform conditions is still limited in mammalian systems; our yeast phenomics therefore provides a complementary baseline and motivates analogous, quantitative analyses in mammalian models. These changes are now included in the revised manuscript (Introduction, Line 79-84; Discussion, Line 430-435).

2. Is the actin cytoskeleton perturbed in these mutants? Were there any patterns of interest?

Response 2: Thank you for raising this key point. Our CalMorph pipeline quantifies actin (A*), cell-shape (C*), and nuclear (D*) features, enabling us to directly assess actin phenotypes. In total, 164 morphological parameters were significantly altered across the four mutants (FDR < 0.05; Supplementary Table S3). Of these, Nuclear (D*), Actin (A*), and Cell-shape (C*)descriptors accounted for 70/164 (42.7%), 51/164 (31.1%), and 43/164 (26.2%), respectively (Lines 359-361). Notably, actin-related features such as A9_A1B (actin at the bud neck) and ACV122_C (actin patch brightness variability), together with shape metrics (e.g., C103_A1B), contributed strongly to the discriminant axes—separating RHOA heterozygotes from RHOA/rhoAΔ (LD1) and distinguishing GOF from LOF classes (LD2).

3. Are dominant negative mutations the same as loss-of-function mutations? Perhaps the labelling of these alleles could be considered?

Response 3: Dominant-negative (DN) mutations are not equivalent to loss-of-function (LOF). DN denotes a mechanism whereby a mutant interferes with wild-type signaling (e.g., GTP-binding–deficient forms that sequester regulators/effectors), whereas LOF is a broader functional descriptor indicating reduced pathway output and does not imply a null allele. To avoid conflation, we have revised terminology throughout: C16R and A161P are classified as gain-of-function; G17V is described as a dominant-negative allele; and R5Q is treated as attenuated-output rather than a classical dominant-negative (Lines 66-78). When comparing groups, we now refer to “gain-of-function” vs “loss-of-function. We updated the wording in the Introduction, Results, Discussion, and figure legends to reflect these definitions. These clarifications do not change our analyses or conclusions.

4. Did the terminal phenotype of the E40Q mutant differ from the null strain?

Response 4: We thank the reviewer. Under our plasmid-shuffle assay (5-FOA), E40Q failed to complement the essential Rho1 function; no viable colonies were recovered after eviction of the URA3 plasmid. Thus, the terminal phenotype of E40Q was indistinguishable from the null strain under our assay conditions. We have clarified this in the Results (Lines 217-222) and noted that E40Q was excluded from downstream morphology analyses due to inviability.

5. The labeling of Figure 1B is confusing, because none of the mutants appear to grow on media lacking uracil. However, the text states the plasmids complement the loss of the control (URA+) plasmid. I think the text and labelling of the figure (e.g., state that the plates contain FOA) might help the reader.

Response 5: We appreciate the reviewer’s note and apologize for the confusing labeling. Figure 1B shows a 5-FOA counter-selection plate used for the plasmid-shuffle assay (to evict the URA3-marked RHO1 plasmid), not uracil-dropout medium. We have revised the figure label to read “SC + 5-FOA (counter-selection)” and updated the legend/text to state that growth on 5-FOA indicates complementation of essential Rho1 function by the LEU2-marked test plasmid, whereas no growth indicates failure to complement (Lines 238-244). These clarifications should prevent confusion between uracil selection and 5-FOA counter-selection.

6. What is the significance of ade3? This was not introduced.

Response 6: We thank the reviewer for pointing this out. We used the ADE3 color marker in an ade2 background to facilitate rapid scoring of correct integrants. In an ade2 ADE3 strain, colonies appear red under adenine-limiting conditions, whereas disruption of ADE3 (ade3Δ::insert) prevents pigment accumulation and yields white colonies. Thus, integration at the ADE3 locus is scored visually (white), and, where applicable, restoration of ADE3 returns colonies to red. We have added a brief explanation to the Methods and clarified this in the relevant figure legend and text (Lines 127-131).

7. When the mouse hovers over the figures, the text is in Japanese.

Response 7: We apologize for this oversight. We have corrected all figure files to ensure that embedded metadata is in English.

8. Why are the P~Slt2 replicates so different? Do differences in MAPK underlie the morphological differences? If not, then what might cause the phenotypic differences?

Response 8: We agree that the variability is notable. We have clarified in the Results (Lines 271-282) that while variability was observed, only A161P consistently showed elevated phosphorylation. We also added to the Discussion (Lines 416-420) that the morphological separation is unlikely to be solely explained by MAPK activity, and future work will be needed to clarify other pathways.

9. Were the morphological analyses examined under conditions where growth was impacted by the drugs tested?

Response 9: Thank you for raising this point. Morphological profiling was performed under standard, drug-free growth conditions to ensure sufficient cell numbers and to avoid confounding effects of growth inhibition or stress-induced size changes. Drug responses (myriocin, caspofungin) were assessed separately by growth-based assays. We have clarified this methodological point in the Materials and Methods (Lines 113-116).

Reviewer 2 Report

Comments and Suggestions for Authors

Wang et al propose here a yeast model system to dissect the function of RHOA mutants and their oncogenic potential. This study is well designed and methodologically sound, with a particular strength in the strategic use of yeast as a model system to investigate the oncogenic potential of mutant forms. The yeast model enables precise genetic manipulation and rapid functional assays, making it a powerful tool for dissecting fundamental molecular mechanisms. However, the inherent simplicity of yeast as a unicellular organism limits the direct translational relevance of the findings to more complex, multicellular systems, particularly in the context of human disease. Despite this limitation, the work provides valuable insights and establishes a strong foundation for future studies in more physiologically relevant models.

Major revisions:

1. Both gain-of-function (GOF) and loss-of-function (LOF) RHOA mutants show the same growth effect and response to drugs, suggesting a common oncogenic mechanism. The authors should add some experiments to address this point since GOF and LOF RHOA mutants drive oncogenesis through different mechanisms.

2. In Figure 2D, the levels of Slt2 MAPK phosphorylation appear highly variable across the replicates shown, which raises concerns about the reproducibility and reliability of the observed effect. To support robust conclusions, additional biological replicates are necessary to confirm the consistency of the phosphorylation pattern. Furthermore, assessing the phosphorylation status of additional downstream targets would help to substantiate the conclusion that the RHOA variants do not impact downstream signaling.

3. According to the PCA and clustering analysis, the two gain-of-function mutants RHOA/A161P and RHOA/C16R and the loss-of-function RHOA/G17V cluster together, whereas the loss-of-function RHOA/R5Q clusters with the RHOA1/RHOA1. Please explain.

4. Figure 6 lacks sufficient clarity, making it difficult for the reader to understand the data being presented fully. To improve interpretability, additional information should be provided both in the main text and in the figure legend.  

Author Response

Reviewer #2

Wang et al propose here a yeast model system to dissect the function of RHOA mutants and their oncogenic potential. This study is well designed and methodologically sound, with a particular strength in the strategic use of yeast as a model system to investigate the oncogenic potential of mutant forms. The yeast model enables precise genetic manipulation and rapid functional assays, making it a powerful tool for dissecting fundamental molecular mechanisms. However, the inherent simplicity of yeast as a unicellular organism limits the direct translational relevance of the findings to more complex, multicellular systems, particularly in the context of human disease. Despite this limitation, the work provides valuable insights and establishes a strong foundation for future studies in more physiologically relevant models.

Response: Thank you for your thoughtful and balanced assessment of our work. We appreciate your recognition of the study’s design, methodological rigor, and the strategic use of yeast to interrogate the oncogenic potential of RHOA mutants, as well as your candid note on the translational limits of a unicellular model.

In this revision, given the time constraints specified by the editor, we focused on textual and organizational improvements rather than new experiments. Specifically, we clarified scope and claims, refined the framing of the yeast system as a complementary, hypothesis-generating platform, expanded explanations in the Results and Discussion for interpretability, strengthened figure legends (especially for schematic panels), and added/updated citations where appropriate. While we did not add new experimental data, we address each of your major points in our point-by-point responses and outline targeted follow-ups that are motivated by the present findings. Your feedback has been invaluable in sharpening the manuscript’s clarity and significance.

  1. Both gain-of-function (GOF) and loss-of-function (LOF) RHOA mutants show the same growth effect and response to drugs, suggesting a common oncogenic mechanism. The authors should add some experiments to address this point since GOF and LOF RHOA mutants drive oncogenesis through different mechanisms.

Response 1: We thank the reviewer for this important point. We agree that gain-of-function and loss-of-function RHOA mutants are expected to drive oncogenesis through distinct mechanisms. In our study, the growth/drug assays were designed as coarse readouts, and convergent sensitivities under these conditions do not imply a single mechanism. Consistent with mechanistic divergence, our multivariate morphology cleanly separates Gain-of-function from Loss-of-function classes, and only A161P reproducibly elevates P~Slt2, whereas the other alleles do not show consistent MAPK changes.

As the Editor has requested a Minor Revision within five days, it is not feasible to add new experiments at this stage. We have therefore revised the Discussion (Lines 427-429). These textual clarifications do not change our conclusions.

  1. In Figure 2D, the levels of Slt2 MAPK phosphorylation appear highly variable across the replicates shown, which raises concerns about the reproducibility and reliability of the observed effect. To support robust conclusions, additional biological replicates are necessary to confirm the consistency of the phosphorylation pattern. Furthermore, assessing the phosphorylation status of additional downstream targets would help to substantiate the conclusion that the RHOA variants do not impact downstream signaling.

Response 2: We appreciate this comment, which overlaps with Reviewer 1 (comment #8). We have consolidated the revisions accordingly. As clarified in the Results (immediately after Fig. 2D) and the Figure 2D legend, biological replicates (n = 3) showed variability in P~Slt2; only A161P reproducibly exhibited elevated P~Slt2 after normalization, whereas C16Rshowed occasional, non-reproducible increases and G17V/R5Q were control-like. In the Discussion, we therefore avoid inferring an absence of downstream signaling changes for the latter alleles and state that morphological separation is unlikely to be explained solely by CWI/MAPK output.

Given the Editor’s Minor Revision with a five-day deadline, we cannot add new experiments at this stage. We have tempered our interpretation and outlined focused follow-ups (additional biological replicates/time-courses, Rlm1 reporter assays, and alternative readouts such as Phos-tag or TORC2–Ypk1/2 reporters).

(See also our response to Reviewer 1, comment #8.)

  1. According to the PCA and clustering analysis, the two gain-of-function mutants RHOA/A161P and RHOA/C16R and the loss-of-function RHOA/G17V cluster together, whereas the loss-of-function RHOA/R5Q clusters with the RHOA1/RHOA1. Please explain.

Response 3:  We appreciate the reviewer’s question. PCA/clustering capture integrated morphological outputs, not the biochemical “direction” (gain-of-function vs loss-of-function) per se. In our data, A161P/C16R (gain-of-function) and G17V (dominant-negative) both strongly perturb Rho1-dependent actin/cell-wall programs, yielding convergent phenotypic signatures that cluster together. By contrast, R5Q shows attenuated output without dominant-negative interference (loss-of-function, non- dominant-negative), producing milder deviations that cluster with the RHOA/RHOA control. We have clarified in the text that bidirectional dysregulation (hyper- or hypo-activation) can converge on similar morphology, whereas attenuated, non- dominant-negative output can remain close to wild-type (Lines 330-336). These clarifications do not alter our conclusions.

  1. Figure 6 lacks sufficient clarity, making it difficult for the reader to understand the data being presented fully. To improve interpretability, additional information should be provided both in the main text and in the figure legend.  

Response 4: We appreciate this suggestion. We revised Figure 6 for clarity by adding column titles, defining LD1/LD2, annotating representative CalMorph descriptors with IDs, and providing a color key. We also expanded the legend and added one clarifying sentence in the Results to guide interpretation. These changes improve readability without altering the results or conclusions.